# Iterated Petrov–Galerkin Method with Regular Pairs for Solving Fredholm Integral Equations of the Second Kind

**Silvia Alejandra Seminara * and María Inés Troparevsky**

Departamento de Matemática, Facultad de Ingeniería, Universidad de Buenos Aires,
C1063ACV Buenos Aires, Argentina; mfiuba@gmail.com
* Correspondence: seminarasilvia@gmail.com; Tel.: +54-911-314-71920

**Abstract:** In this work we obtain approximate solutions for Fredholm integral equations of the second kind by means of Petrov–Galerkin method, choosing "regular pairs" of subspaces, $\{X_n, Y_n\}$, which are simply characterized by the positive definitiveness of a correlation matrix. This choice guarantees the solvability and numerical stability of the approximation scheme in an easy way, and the selection of orthogonal basis for the subspaces make the calculations quite simple. Afterwards, we explore an interesting phenomenon called "superconvergence", observed in the 1970s by Sloan: once the approximations $u_n \in X_n$ to the solution of the operator equation $u - Ku = g$ are obtained, the convergence can be notably improved by means of an iteration of the method, $u_n^* = g + Ku_n$. We illustrate both procedures of approximation by means of two numerical examples: one for a continuous kernel, and the other for a weakly singular one.

**Keywords:** Fredholm integral equations; numerical solutions; Petrov–Galerkin method; regular pairs; iterated methods

---

## 1. Introduction

Fredholm equations of the second kind are integral equations of the form

$$u(t) - \int_a^b k(t,s)u(s)ds = g(t) \ \ \forall t \in [a,b] \tag{1}$$

with $u$ an unknown function in a Banach space $X$. The kernel $k : [a,b] \times [a,b] \to R$ and the right-hand side $g : [a,b] \to R$ are given functions.

They appear in different areas of applied mathematics, sometimes as equivalent formulation for boundary value problems with ordinary differential equations, and there are many problems of mathematical physics that are modelled with Fredholm integral equations with different kernels (see, for example, [1–3]).

The equation may be written

$$u - Ku = g \tag{2}$$

by defining the operator $K : X \to X$, $K(u)(.) = \int_a^b k(.,s)u(s)ds$.

If for the kernel $k(t,s)$ the operator $K$ is bounded, a sufficient condition to guarantee the existence and uniqueness of a solution of Equation (2) is that $\|K\| < 1$ (see [4], Theorem 2.14, p. 23).

Petrov–Galerkin is a projection method often proposed to find numerical approximate solutions to this type of integral equation. The idea is to choose appropriate sequences of finite dimensional subspaces of $X$, $\{X_n\}_{n \in N}$ and $\{Y_n\}_{n \in N}$, the trial and test subspaces respectively, where the unknown $u$ and the data $g$ are to be projected.

In the case of being $X$ a Hilbert space with inner product $\langle .,. \rangle$, the Petrov–Galerkin method looks for $u_n \in X_n$ such that

$$\langle u_n - Ku_n, v \rangle = \langle g, v \rangle \quad \forall v \in Y_n \tag{3}$$

and, as $X_n$ and $Y_n$ are subspaces of dimension $d_n < \infty$, solving Equation (3) reduces to solve a linear algebraic system of equations represented by a $d_n \times d_n$ matrix.

In [5] it is proved that, if $K : X \to X$ is a compact linear operator not having 1 as an eigenvalue, and the pair $\{X_n, Y_n\}$ is a regular pair—a concept to be defined in the next section—Equation (3) has a unique solution $u_n$ which satisfies

$$\|u - u_n\|_X \leq C.inf_{x \in X_n}\|x - u\|_X \tag{4}$$

where the constant C does not depend on $n$.

Solvability and numerical stability of the approximation scheme are, in this way, assured, and the accuracy of the approximation $u_n$ to the unique solution $u$ of Equation (2) does not depend formally on $Y_n$, as can be noted in Equation (4). The goal is to choose test function subspaces $Y_n$ that are easy to handle, while the quality of convergence of the method is preserved.

In addition, the convergence can even be improved by means of an iteration of the method: once the approximations $u_n \in X_n$ are obtained, a new sequence of approximate solutions $u_n^* \in X_n$ can be built by means of a simple procedure (see [6–8]):

$$u_n^* = g + Ku_n \tag{5}$$

In this work we choose pairs of simple subspaces $\{X_n, Y_n\}$ generated by Legendre polynomials and show the goodness of the approximations in two numerical examples with known solution, one of them having singular kernel. We then improved the convergence by means of an iteration of the method and show why the approximation is better, even for small values of $n \in N$.

## 2. Method

Let $(X, \langle .,. \rangle)$ be a Hilbert space, $\|.\|$ the associated norm, and $K : X \to X$ a compact linear operator. It is shown in [4] that, if $\|K\| < 1$, there exists a solution $u \in X$ to Equation (2) for $g \in X$ a given function, and it is unique. We are interested in looking for a good approximation to $u \in X$ satisfying Equation (2).

For each $n \in N_0$ let us consider subspaces $X_n \subset X, Y_n \subset X$, with $\dim(X_n) = \dim(Y_n) = d_n < \infty$. The Petrov–Galerkin method for Equation (2) is a numerical method to find $u_n \in X_n$ satisfying Equation (3).

For the method to be useful, it is necessary to establish conditions under which Equation (3) has a unique solution $u_n \in X_n$ and $\lim_{n\to\infty}\|u - u_n\|_X = 0$ for $u$ the unique solution of Equation (2).

It is easy to show that the condition

$$X_n{}^{\perp} \cap Y_n = \{0\} \tag{6}$$

ensures the existence of a unique solution $u_n \in X_n$ for Equation (3). From [4] (p. 243), convergence can be expected only if

$$\forall x \in X, \exists\{x_n, n \in N\} \subset X_n : \lim_{n\to\infty} x_n = x \tag{7}$$

so, from now on, the sequences of subspaces $\{X_n\}_{n\in N}$ and $\{Y_n\}_{n\in N}$ are both chosen verifying this condition of denseness.

Following [5], and denoting by $\{X_n, Y_n\}$ the sequences of subspaces, the pair $\{X_n, Y_n\}$ is said to be "regular" if there exists a linear surjective operator $\Pi_n : X_n \to Y_n$ satisfying

i.  $\exists C_1 \in R / \forall x \in X_n : \|x\| \leq C_1 \sqrt{\langle x, \Pi_n x \rangle}$

ii.  $\exists C_2 \in R / \forall x \in X_n : \|\Pi_n x\| \leq C_2 \|x\|$

It is easy to show that the surjectivity of $\Pi_n$ and (i.) assure the condition of Equation (6).

From [5] (p. 411), the following theorem summarizes the conditions for the existence and uniqueness of the solutions of Equation (3) and their convergence to the solution of Equation (2):

**Theorem 1.** *Let $X$ be a Hilbert space and $K : X \to X$ a compact linear operator not having 1 as eigenvalue. Suppose $X_n$ and $Y_n$ are finite dimensional subspaces of $X$, with $\dim(X_n) = \dim(Y_n)$, verifying that $\{X_n, Y_n\}$ is a regular pair and, for each $x \in X$, there exist sequences $\{x_n, n \in N\} \subset X_n$ and $\{y_n, n \in N\} \subset Y_n$ so that $\lim_{n \to \infty} x_n = x$ and $\lim_{n \to \infty} y_n = x$. Then, there exists $n_0 \in N$ such that, for $n > n_0$, equation $\langle u_n - Ku_n, v \rangle = \langle g, v \rangle \; \forall v \in Y_n$ has a unique solution $u_n \in X_n$ for any given $g \in X$, that satisfies $\|u - u_n\| \leq C.inf_{x \in X_n} \|x - u\|$, where $u \in X$ is the unique solution of $u - Ku = g$ and $C$ is constant not dependent of $n$.*

From [9], the characterization of a regular pair is simple by means of the so called "correlation matrix". Let $\{\varphi_i^n, i = 1, \ldots, d_n\}$ and $\{\psi_j^n, j = 1, \ldots, d_n\}$ be bases of $X_n$ and $Y_n$, respectively, and define the $d_n \times d_n$ matrices $[G(X_n)]_{ij} := \langle \varphi_i^n, \varphi_j^n \rangle$, $[G(Y_n)]_{ij} := \langle \psi_i^n, \psi_j^n \rangle$, the correlation matrix $[G(X_n, Y_n)]_{ij} := \langle \varphi_i^n, \psi_j^n \rangle$, and $[G_+(X_n, Y_n)]_{ij} = \frac{1}{2}([G(X_n, Y_n)]_{ij} + [G(X_n, Y_n)]_{ji})$, for $i, j = 1, \ldots, d_n$.

Note that for the real case, $G(X_n)$ and $G(Y_n)$ are positive definite and $G_+(X_n, Y_n)$ is the symmetric part of the correlation matrix. We have proven (see [10]) the following

**Proposition 1.** *If $G_+(X_n, Y_n)$ is positive definite, $\{X_n, Y_n\}$ is a regular pair.*

For conciseness, we assume $[a, b] = [0, 1]$ from here on.

Let us consider $X = L^2([0, 1])$.

For the interval $[0, 1]$, $S_n^m$ is the subspace of polynomials of degree less than $m$ on each subinterval $I_{j,n} = (\frac{j}{2^n}, \frac{j+1}{2^n}), j = 0, 1, \ldots, 2^n - 1$; $\dim(S_n^m) = m2^n$ and $S_0^m \subset S_1^m \ldots \subset \cup_{n=0}^\infty S_n^m S^m$. $\overline{S^m} = L^2([0, 1])$ since every continuous functions with compact support on $[0, 1]$ can be approximated by steps functions on subintervals of the form $(\frac{j}{2^n}, \frac{j+1}{2^n}), j = 0, 1, \ldots, 2^n - 1$, and they are dense in $L^2([0, 1])$. The condition of Equation (7) of denseness is, so, satisfied.

As the basis of $S_n^m$, we will choose Legendre polynomials of degree less than $m$, adapted to each of the subintervals $I_{j,n}$: $S_n^m = \text{span}\{p_i^{j,n}, j = 0, 1, \ldots, 2^n - 1, i = 0, 1, \ldots, m - 1\}$ with $p_i^{j,n}(x) = Q_i^{j,n}(x) / \|Q_i^{j,n}(x)\|$, $Q_i^{j,n}(x) = L_i(2^n(2x - \frac{2j+1}{2^n})).\chi_{I_{j,n}}$, $L_i(x)$ the Legendre polynomial of degree $i$ on $[-1, 1]$ and $\chi_{I_{j,n}}$ the characteristic function of the subinterval $I_{j,n}$.

We rename $q_l^{i,n} := p_l^{i,n+1}$ to simplify the notation and choose the sequences of subspaces $X_n = S_n^2 = \text{span}\{p_0^{0,n}, p_1^{0,n}, p_0^{1,n}, p_1^{1,n}, \ldots, p_0^{2^n-1,n}, p_1^{2^n-1,n}\}$ and $Y_n = S_{n+1}^1 = \text{span}\{q_0^{0,n}, q_0^{1,n}, \ldots, q_0^{2^{n+1}-1,n}\}$, with $\dim(X_n) = 2.2^n = 2^{n+1} = 1.2^{n+1} = \dim(Y_n) = d_n$.

Note that the condition of Equation (6), $X_n^\perp \cap Y_n = \{0\}$, which assures the uniqueness of the solution of Equation (3) for each $n$, is fulfilled.

Indeed, suppose that $q_0^{j,n} \in Y_n$, for $j$ between 0 and $2^{n+1} - 1$, satisfies that $q_0^{j,n} \perp p_0^{i,n}$ and $q_0^{j,n} \perp p_1^{i,n}$ for every $i = 0, \ldots, 2^n - 1$; then $\int_{\frac{j}{2^{n+1}}}^{\frac{j+1}{2^{n+1}}} q_0^{j,n}.p_0^{\frac{j}{2},n} dx = \frac{q_0^{j,n}.p_0^{\frac{j}{2},n}}{2^{n+1}} = 0$ if $j$ is even or $\int_{\frac{j+1}{2^{n+1}}}^{\frac{j+2}{2^{n+1}}} q_0^{j+1,n}.p_0^{\frac{j+1}{2},n} dx = $

$\frac{q_0^{j+1,n}.p_0^{\frac{j+1}{2},n}}{2^{n+1}} = 0$ if $j$ is odd, which is impossible, since $q_0^{j,n} \neq 0$ and $p_0^{i,n} \neq 0$ for every $j$ and every $i$.

Renaming the elements of the basis as $\varphi_i^n(x) := p_0^{(i-1)2^{-1},n}$ for $i$ odd, $\varphi_i^n(x) := p_1^{(i-2)2^{-1},n}(x)$ for $i$ even and $\psi_j^n(x) := q_0^{j,n}$, it is easy to show that $\{X_n, Y_n\}$ is a regular pair, since $G_+(X_n, Y_n)$ is a $2^{n+1} \times 2^{n+1}$ matrix with definite positive $2 \times 2$ blocks on its principal diagonal and 0s everywhere else (for details, see [10]).

Once the approximations $u_n$ are obtained, an almost natural iteration procedure is possible to obtain new approximations of the real solution $u$. Since the equation being solved is $u - Ku = g$, or $u = g + Ku$, we can define $u_n^* = g + Ku_n$. This first iteration, applied to the Galerkin method, has been studied since the 1970s, because, under appropriated conditions of $K$ and $g$, it reveals an interesting phenomenon called "superconvergence" (see [6,11], for instance), as the order of convergence can be notably improved.

In [11] (p. 42), the existence of a unique solution $u_n^*$ for Equation (5) and the improvement of the order of convergence of the iterated approximation for any projection method are guaranteed.

In [5] (p. 419), the superconvergence in the Petrov–Galerkin scheme applied to Fredholm equations of the second kind is explained and, under the same conditions of the Theorem 1 we have just enunciated, a theorem establishes that $u_n^*$ satisfies

$$\|u - u_n^*\|_2 \leq C.\text{ess sup}_{s \in [0,1]} \left[ \inf_{\psi \in Y_n} \|k(.,s) - \psi\|_2 \right].\inf_{x \in X_n} \|x - u\|_2 \tag{8}$$

for $u$ the unique solution in $L^2([0,1])$ of Equation (1), showing that the improvement of the order of convergence by the iteration procedure is due to the approximation of the kernel $k$ by elements $\psi$ of test subspace $Y_n$. In our work, the elements of test subspaces $Y_n = S_{n+1}^1$ are piecewise constant functions on the dyadic subintervals $I_{j,n} = \left( \frac{j}{2^n}, \frac{j+1}{2^n} \right), j = 0, 1, \ldots, 2^n - 1$.

We will now follow an idea from [6] (p. 67). For $f$ a Lipschitz function on an interval $I$, with Lipschitz constant $L$, let $\psi_{\frac{1}{2}}$ be the piecewise constant function defined by $\psi_{\frac{1}{2}}(t) = f(t_i + \frac{h}{2})$ for $t \in I_i = [t_i, t_i + h]$, with $I = \cup I_i$ a regular partition of $I$ with norm $h$.

For any $I_i$, if $t \in I_i : |f(t) - \psi_{\frac{1}{2}}(t)| \leq \frac{hL}{2}$, so $\|f - \psi_{\frac{1}{2}}\|_\infty \leq \frac{hL}{2}$.

If the kernel $k$ satisfies that $k(.,s) = k_s(.)$ is a Lipschitz function with Lipschitz constant $L_s$ for each $s \in [0,1]$, it is $\|k_s - \psi_{\frac{1}{2}}\|_\infty \leq \frac{1}{2}\frac{1}{2^{n+1}}L_s$ and, then, $\inf_{\psi \in Y_n}\|k_s - \psi\|_\infty \leq \frac{L_s}{2^{n+2}}$ for each $s \in [0,1]$ and, consequently, $\text{ess sup}_{s \in [0,1]} \left[ \inf_{\psi \in Y_n} \|k(.,s) - \psi\|_2 \right] \leq \frac{1}{2^{n+2}}\text{ess sup}_{s \in [0,1]} L_s$.

Moreover, if $\text{ess sup}_{s \in [0,1]} L_s < \infty$, from Equation (8), $\|u - u_n^*\|_2 \leq C.\frac{1}{2^{n+2}}\text{ess sup}_{s \in [0,1]} L_s.\inf_{x \in X_n} \|x - u\|_2$, and the approximation is actually improved.

## 3. Results

We will offer two numerical examples of the goodness of the Petrov–Galerkin method and iterated Petrov–Galerkin method with regular pairs, applied to Fredholm integral equations of the second kind: one with a continuous kernel, and the other with a weakly singular kernel ([4], p. 29; [12], p.7).

The kernel $k : [a, b] \times [a, b] \rightarrow R$ is said to be weakly singular if it verifies

$$|k(s,t)| \leq M|s - t|^{\alpha - 1} \quad \forall (s,t) \in [a,b] \times [a,b], s \neq t \tag{9}$$

with $0 < \alpha < 1$ and $M \in R$.

Both for $k$ a continuous kernel or a weakly singular one, $K : L^2([0,1]) \rightarrow L^2([0,1])$ is compact operator (see [4], p. 28, Theorem 2.28; and [13], p. 582, Theorem 1, respectively).

We have chosen "regular pairs" of subspaces, and orthogonal basis for them, reducing the difficulty of calculations.

We worked with $X_n = S_n^2 = \text{span}\{\varphi_i^n, i = 1, \ldots, 2^{n+1}\}$ and $Y_n = S_{n+1}^1 = \text{span}\{\psi_j^n, j = 1, \ldots, 2^{n+1}\}$, with $\varphi_i^n(x) = \sqrt{2^n}.\chi_{I_{i-1,n}}$ for $i$ odd, $\varphi_i^n(x) = \sqrt{3.2^n}(2^{n+1}x - i + 1).\chi_{I_{i-1,n}}$ for $i$ even and $\psi_j^n(x) = \sqrt{2^{n+1}}.\chi_{I_{j-1,n+1}}$.

Note that the trial space $X_n$ is generated by piecewise constant and piecewise linear orthogonal functions; in [14], only piecewise linear (not orthogonal) functions are used.

### *3.1. Numerical Examples*

#### 3.1.1. Example 1

The equation

$$u(t) - \frac{1}{2}\int_0^1 e^{st}u(s)ds = g(t) \quad \forall t[0,1] \tag{10}$$

with $g(t) = te^t - \frac{1}{2}(1 + te^{t+1})(1 + t)^{-2}$, has the exact solution $u(t) = te^t$.

The linear operator $K : L^2([0,1]) \to L^2([0,1])$, $K[u](t) = \frac{1}{2}\int_0^1 e^{st}u(s)ds$ is compact because $k(t,s) = e^{st}$ is continuous, and $\|K\| \leq (\int_0^1(\int_0^1 k^2(t,s)ds)dt)^{\frac{1}{2}} < 1$, thus 1 is not an eigenvalue of $K$ and convergence of the Petrov–Galerkin method to the (unique) exact solution is guaranteed.

In Figure 1a we plot the exact solution together with the approximations $u_0, u_1, u_2$ and $u_3$. The quadratic errors with respect to the exact solution $u$, $\varepsilon_n = \|u - u_n\|_2$, are, respectively, $\varepsilon_0 \sim 0.160157$, $\varepsilon_1 \sim 0.043244$, $\varepsilon_2 \sim 0.011036$ and $\varepsilon_3 \sim 0.002773$.

In Figure 1b, the plots of the exact solution together with $u_1^*, u_2^*$ and $u_3^*$ are shown. The quadratic errors with respect to $te^t$ are, in this case, $\varepsilon_0^* \sim 0.005657$, $\varepsilon_1^* \sim 0.000849$ and $\varepsilon_2^* \sim 0.000139$.

Note that the plots of the iterated approximations and the real solution are indistinguishable.

All the approximations were obtained by means of ad hoc designed algorithms, implemented with Wolfram Mathematica® 9.

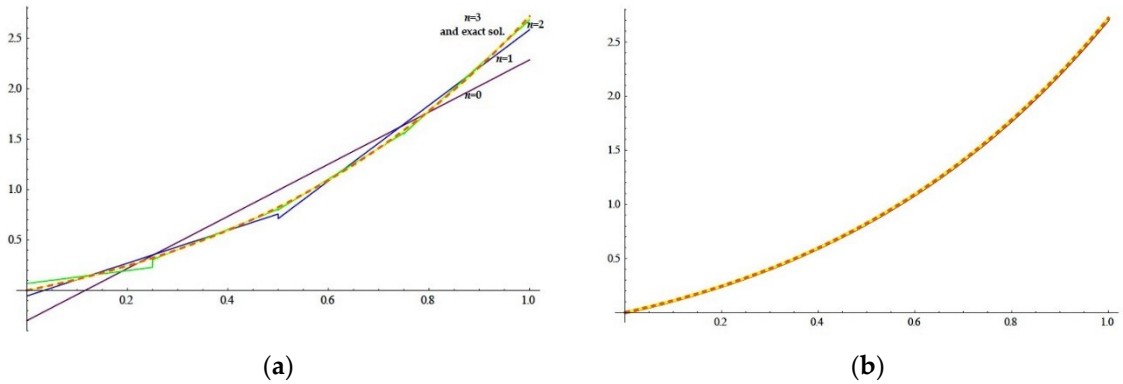

|(**a**)|(**b**)|

**Figure 1.** (**a**) Approximations to exact solution of Equation (10) before iteration: Purple for $n = 0$, blue for $n = 1$, green for $n = 2$, yellow for $n = 3$, and, dashed in red, the exact solution. The quadratic errors with respect to the exact solution are, respectively, $\varepsilon_0 \sim 0.160157$, $\varepsilon_1 \sim 0.043244$, $\varepsilon_2 \sim 0.011036$ and $\varepsilon_3 \sim 0.002773$; (**b**) the approximations for $n = 1$, 2 and 3 after iteration are graphically indistinguishable from the exact solution of the equation (10). The quadratic errors with respect to $te^t$ are, in this case, $\varepsilon_1^* \sim 0.005657$, $\varepsilon_2^* \sim 0.000849$ and $\varepsilon_3^* \sim 0.000139$.

#### 3.1.2. Example 2

The equation

$$u(t) - \frac{1}{\sqrt{3}}\int_0^1 \frac{u(s)}{\sqrt[4]{|t - s|}}ds = g(t) \quad \forall t \in [0,1] \tag{11}$$

with $g(t) = t^2 - t^3 - \frac{4}{231\sqrt{3}}(32t^{\frac{11}{4}} + (1 - t)^{\frac{3}{4}}(21 + 24t + 32t^2) + \frac{128}{5}t^{\frac{15}{4}} + \frac{(1-t)^{\frac{3}{4}}}{5}(77 + 84t + 96t^2 + 128t^3))$, has the exact solution $u(t) = t^2 - t^3$.

The kernel $k(t,s) = \frac{1}{\sqrt{3}}\frac{1}{\sqrt[4]{|t-s|}}$ is weakly singular, with $\alpha = \frac{3}{4}$, according to (9).

Theorem 1 from [13] (p. 582) guarantees the compactness of the operator $K : L^2([0,1]) \to L^2([0,1])$, $K[u](t) = \frac{1}{\sqrt{3}} \int_0^1 \frac{u(s)}{\sqrt[4]{|t-s|}} ds$, because the necessary and sufficient conditions are verified: $\sup_{t \in [0,1]} \|k(t,.)\|_2 = \sup_{t \in [0,1]} \left( \frac{1}{3} \int_0^1 \frac{ds}{\sqrt{|t-s|}} \right)^{\frac{1}{2}} \le \frac{2}{\sqrt{3}} < \infty$ and $\lim_{t \to \tau} \|k(t,.) - k(\tau,.)\|_2 = 0$ for $\tau \in [0,1]$.

It is $\|K\| \le \left( \int_0^1 (\int_0^1 k^2(t,s)ds)dt \right)^{\frac{1}{2}} < 1$, thus 1 is not an eigenvalue of $K$ and convergence of the method to the (unique) exact solution is guaranteed.

In Figure 2a we plot the exact solution together with the approximations $u_0, u_1, u_2, u_3$ and $u_4$ obtained with Mathematica®, and in Figure 2b, the exact solution together with $u_0^*, u_1^*$ and $u_2^*$, the last one being practically indistinguishable from the exact solution. By comparing quadratic errors, the improvement of the approximation can be appreciated: for $n = 2$, $\varepsilon_2 = \|u - u_2\|_2 < 0.0046$ and $\varepsilon_2^* = \|u - u_2^*\|_2 < 0.00023$.

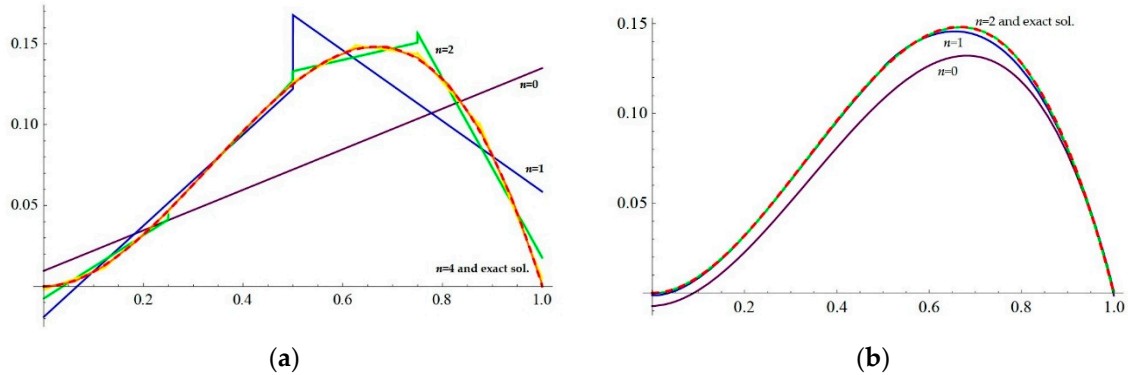

(a)  (b)

**Figure 2.** (**a**) Approximations to exact solution of Equation (11) before iteration: purple for $n = 0$, blue for $n = 1$, green for $n = 2$, yellow for $n = 3$, orange for $n = 4$, and, dashed in red, the exact solution; (**b**) the same approximations after the iteration. For $n = 2$, the approximation is graphically indistinguishable from the exact solution of the Equation (11), and the quadratic errors is reduced from $\varepsilon_2 = \|u - u_2\|_2 < 0.0046$ to $\varepsilon_2^* = \|u - u_2^*\|_2 < 0.00023$.

## 4. Discussion

The Petrov–Galerkin method is applied by choosing appropriate subspaces for projecting. The choice of a "regular pair" of subspaces (easily characterized by the positive definitiveness of a correlation matrix), and orthogonal basis for them, reduce the difficulty of calculations. Iteration is shown to be a very simple way for improving convergence in a remarkable way, and better orders of convergence can be shown, even for a weakly singular kernel. It is necessary to say that, in this second numerical example, we have had difficulties with the fluid implementation of the computational algorithms because of the improper integrals involved. However, not so many computations were necessary since with $n = 2$ we have obtained very good results. In [14,15], the authors propose discrete methods to face the numerical difficulties arising from the calculation of improper integrals involved in the case of weakly singular kernel.

It is appropriate to point out that, in recent papers, different discrete Galerkin approaches were proposed to solve integral equations. In particular, meshless discrete Galerkin methods were successfully developed for solving Fredholm and Hammerstein integral equations for various bases. See, for example, [16] for an effective and stable method to estimate the solution to Hammerstein integral equations with free shape parameter radial basis functions, constructed on scattered points; [17,18], for effective computational meshless methods for solving Fredholm integral equations of the second kind with logarithmic and weakly singular kernels, using radial basis functions, meshless product integration and collocation methods; and [19,20], for efficient meshless methods for solving

non-linear weakly singular Fredholm integral equations, combining discrete collocation method with locally supported radial basis functions and thin-plate splines.

Finally, a plausible line for our future work could be to explore and take advantages of some of these discrete methods of approximation to avoid the difficulties of calculations arising from the improper integrals when solving Fredholm integral equations of the second kind with weakly singular kernels.

**Author Contributions:** Conceptualization, M.I.T.; formal analysis, S.A.S. and M.I.T.; investigation, S.A.S. and M.I.T.; methodology, M.I.T.; project administration, M.I.T.; software, S.A.S. and M.I.T.; supervision, M.I.T.; validation, S.A.S. and M.I.T.; visualization, S.A.S.; writing—original draft, S.A.S.; writing—review and editing, S.A.S. and M.I.T.

**Funding:** This research was partially supported by Universidad de Buenos Aires, UBACyT 2018-2021, 20020170100350BA.

**Conflicts of Interest:** The authors declare no conflict of interest.

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
