# Peer review of "Iterated Petrov–Galerkin Method with Regular Pairs for Solving Fredholm Integral Equations of the Second Kind"

_mca, doi:10.3390/mca23040073_

Round 1

Reviewer 1 Report

Report on the submitted paper in "Mathematical and Computational Applications" entitled

"Iterated Petrov-Galerkin Method with Regular Pairs for Solving Fredholm Integral Equations of the Second Kind"

by Silvia Alejandra Seminara and Maria Inws Troparevsky

Manuscript Number: mca-384600

This paper investigates a computational method for solving linear Fredholm integral equations of the second kind. The proposed scheme in the current work is based on the use of iterated Petrov-Galerkin method by projecting on appropriate finite dimensional subspaces. The authors claim that choosing regular pairs of subspaces guarantees easily the solvability and numerical stability of the presented scheme. The authors given two numerical examples, including integral equations with continuous and weakly singular, to show the validity of the technique and compared the results with the exact solution (simple subspaces  generated by Legendre polynomials). This paper seems to be scientifically correct, well-written and interesting. However, I recommend it be accepted for publication after the authors resolve the some required revisions.

In the following, the required revisions are listed:

1.The authors can be given more commented about the comparison between the presented scheme with the other method.  In fact, they can point out in details some advantages of presented method beside another methods for solving Fredholm integral equations.

2. Introduction should be extended over some applications of integral equations of the second kind, the following references can be useful:

[A meshless discrete Galerkin (MDG) method for the numerical solution of integral equations with logarithmic kernels, Journal of Computational and Applied Mathematics, 267 (2014) 160-181]

[The numerical solution of weakly singular integral equations based on the meshless product integration (MPI) method with error analysis, Applied Numerical Mathematics, 81(2014)76–93]

3. The authors have not discussed on the information about computer and the mathematical software which has been used in the current paper. Some details about these should be consider.

4. I would like the authors to contrast their results with other numerical solution methods (if possible).

5. To show the better understanding of the effects observed in the Figures in the numerical results, the author can explain some additional notes about these.

6. The recent works for solving integral equations using meshless methods could be cited in the introduction of the new version with some comments, such as

[The numerical solution of two.dimensional logarithmic integral equations on normal domains using radial basis functions with polynomial precision, Engineering with Computers, 33:853-870, 2017.]

[Thin plate spline Galerkin scheme for numerically solving nonlinear weakly singular Fredholm integral equations, Applicable Analysis, 2018, DOI:10.1080/00036811.2018.1448073]

[On the numerical solution of Fredholm integral equations utilizing the local
radial basis function method, International Journal of Computer Mathematics, 2018, 10.1080/00207160.2018.1500693]

[A Meshless Discrete Galerkin Method Based on the Free Shape Parameter Radial Basis Functions for Solving Hammerstein Integral Equations, Numer. Math. Theor. Meth. Appl., 2018, Vol. 11, No. 3, pp. 541-569]

7. How can the method be applied to other integral equation? Some details about this subject could be provided (such as nonlinear and Volterra integral equations).

8. Generally, English and presentations are acceptable, but some English and statements should be
clarified and improved for publications.

Author Response

Response to Reviewer 1 Comments

Point 1: The authors can be given more commented about the comparison between the presented scheme with the other method.  In fact, they can point out in details some advantages of presented method beside another methods for solving Fredholm integral equations.

Response 1: We have pointed out that the orthogonality of the selected basis makes the calculations easier (it was not explicit in the previous version). Another advantages have also been mentioned: the simplicity of the procedure to determine if a pair of subspaces is a regular one, and the fact that the accuracy of the approximation doesn’t depend on the subspaces Yn, so they can be chosen conveniently, to make the calculations simpler.  

Point 2: Introduction should be extended over some applications of integral equations of the second kind.

Response 2: We have added some references related to the applications (thanks very much for your suggestions).

Point 3: The authors have not discussed on the information about computer and the mathematical software which has been used in the current paper. Some details about these should be consider.

Response 3: We have added some information about these tools., thank you. 

Point 4: I would like the authors to contrast their results with other numerical solution methods (if possible).

Response 4: We are very sorry but it was not possible for us to test other methods in the short time we have to answer your kind suggestions.  It could be the subject for a future work, because we haven’t compared the method with other ones for the moment, and it is an important and interesting matter to work about.

Point 5: To show the better understanding of the effects observed in the Figures in the numerical results, the author can explain some additional notes about these.

Response 5: We have added some information about the approximations (quadratic errors, for example).

Point 6:  The recent works for solving integral equations using meshless methods could be cited in the introduction of the new version with some comments

Response 6: We have added some comments about these methods in our discussion, because we found them extremely adequate to avoid some problems we have had with the improper integrals in the case of weakly singular kernels.  Thank you very much for your kind and timely suggestions.

Point 7: How can the method be applied to other integral equation? Some details about this subject could be provided (such as nonlinear and Volterra integral equations).

Response 7: This is another point we couldn’t answer in this short time: we are very sorry. We consider it is a subject that deserves a detailed study and we will take it into consideration in future lines of our work.  

Point 8:  Generally, English and presentations are acceptable, but some English and statements should be clarified and improved for publications.

Response 8: We have made some modifications.  We hope our English has improved, even if it's a little…

We are very grateful for every of your useful and pertinent suggestions.

Reviewer 2 Report

As general/major suggestions, the reviewer recommends the authors to review the abstract (it looks rather like an introduction than an abstract) and include therein a sentence about the orthogonality of the basis functions. Moreover, there are formulas that should be written in equation mode rather than within regular text (one of those is in L162-163). The claim that Condition (6) is satisfied (L103-104) does not seem obvious, as spaces Xn and Yn employ different domain discretizations. Please provide a proof for that claim. Finally, the authors should add a remark (either in Section 2 or in Section 3) that their approximation spaces are the same as in reference [11] (or aren't they?) but the basis functions are different (in fact, orthogonality is the key difference).

Detailed corrections follow:

L15:  of equations -> of equation

L17: election -> choice

L31: please remove "and partial" or cite a reference with an example of it (or are you thinking about 2D-3D Fredholm equations?)

L37:  of equations -> of equation

L51: replace "good convergence" with a sharper claim

L61-63: remove "It is shown ...satisfying (2)"  (this has been said)

L64: let consider -> let us consider

L86: a "{" is missing before \varphi_i^n

L86: be basis -> be bases

L88: for X_ni -> for i

L92: insert the sentence "For conciseness, we assume $[a,b]=[0,1]$ from here on"

L92: let consider -> let us consider

L95: steps -> step

L97-100: replace "h" with "l" (note that "h" is used with a different meaning in the next page)

L98: remove "and normalized"

L99: do not divide L_h by its norm and replace "the Legendre" with "the normalized Legendre"

L105-106: replace "Renaming ... q_0^j,n" with the paragraph in L143-144, which is much more clear.

L125: be is -> be

L126: norm -> length

L128: replace "In ... that" with "If the kernel $k$ satisfies"

L132-133: replace "Finally...L_s, and" with "Moreover, if"

L139-L140: move to Section 3.1.2

L141-144: please remove these lines (L141-142 is repeated in the next subsections, and L143-144 should replace L105-106). They may be replaced with a comparison with the basis functions from [11]. The authors could also comment how integrals are evaluated.

L200: "SIAM" is missing

Author Response

Response to Reviewer 2 Comments

Point 1: As general/major suggestions, the reviewer recommends the authors to review the abstract (it looks rather like an introduction than an abstract) and include therein a sentence about the orthogonality of the basis functions.

Response 1: We have rewritten the abstract almost completely, and we have mentioned explicitly the orthogonality of the selected basis and the consequent advantages.

Point 2: Moreover, there are formulas that should be written in equation mode rather than within regular text (one of those is in L162-163).

Response 2: We have corrected the formulas, thank you.

Point 3: The claim that Condition (6) is satisfied (L103-104) does not seem obvious, as spaces Xn and Yn employ different domain discretizations. Please provide a proof for that claim.

Response 3: We have included the proof.

Point 4: Finally, the authors should add a remark (either in Section 2 or in Section 3) that their approximation spaces are the same as in reference [11] (or aren't they?) but the basis functions are different (in fact, orthogonality is the key difference).

Response 4: We added a comment about this subject, thank you.

We have appreciated so much your kind and useful comments, specially the detailed corrections.